# THERE ARE FREE LUNCHES

## ABSTRACT

No-Free-Lunch Theorems state that the performance of all algorithms is the same when averaged over all possible tasks. It has been argued that the necessary conditions for NFL are too restrictive to be found in practice. There must be some information for a set of tasks that ensures some algorithms perform better than others. In this paper we propose a novel idea, "There are free lunches" (TAFL) Theorem, which states that some algorithms can achieve the best performance in all possible tasks, in the condition that tasks are given in a specific order. Furthermore, we point out that with the number of solved tasks increasing, the difficulty of solving a new task decreases. We also present an example to explain how to combine the proposed theorem and the existing supervised learning algorithms.

## 1 INTRODUCTION

No-Free-Lunch (NFL) Theorems Wolpert & Macready (1997) Wolpert (2002) state that "all optimization algorithms perform equally well when their performance is averaged across all possible problems". Some researchers Alabert et al. (2015) argued that the necessary conditions for NFL are too restrictive. In practice, there must be some algorithms that perform better than others. However, no one doubt that no algorithm can optimally solve all tasks.

The implications of the NFL Theorems are very disappointing, as an universal algorithm is a dream in the field of Artificial Intelligence (AI). This paper proposes the "There are free lunches" (TAFL) Theorem. The TAFL Theorem states that some algorithms are optimal for all tasks, in the condition that tasks are given in a specific order. We also conclude that the difficulty of solving a new task can decrease when the number of solved tasks increases.

The TAFL Theorem implies a very different way of solving AI problems compared to the NFL Theorems. It is not a matter that algorithms at our hands are not powerful or suitable enough for a specific task. One can use an algorithm of limited intelligence to solve a complex problems step by step.

We introduce the necessary notation in Section 2 and provide the TAFL Theorem in Section 3. We analyse the task order mentioned in the TAFL Theorem and point out we can reduce calls of learning algorithms in Section 4. An application of the TAFL Theorem is provided in Section 5. Finally, we discuss the implications of the TAFL Theorem in Section 6 and conclude the paper in 7.

## 2 PRELIMINARIES

We restrict attention to supervised learning. But the discussion is also applicable for optimization and search fields.

The entire sample space is denoted as $D$. The distribution space is denoted as $F$. A distribution from $F$ is a mapping $f : \boldsymbol{x} \to \boldsymbol{y}$. A sampling strategy $S$ samples a set of training samples from $f$, denoted as $\boldsymbol{d} = \{(\boldsymbol{x}_0, \boldsymbol{y}_0), ..., (\boldsymbol{x}_m, \boldsymbol{y}_m)\}, \boldsymbol{d} \in D$. We consider the sampling strategy is unique for all tasks.

A supervised learning algorithm generates a mapping (or called a hypothesis) $h : \boldsymbol{x} \to \boldsymbol{y}$ to approximate the distribution $f$, in terms of off-training-set error. The training samples $\boldsymbol{d}$ are provided to an algorithm $\mathfrak{L}_a$ to generate a set of $h$. The possibility of the algorithm $\mathfrak{L}_a$ generating a specific $h$ on $\boldsymbol{d}$ is denoted as $P(h|\boldsymbol{d}, \mathfrak{L}_a)$.

No-Free-Lunch Theorems state the total off-training-set error across all possible $f$ is always the same for arbitrary algorithms $\mathfrak{L}_a$ and $\mathfrak{L}_b$.

$$\sum_f E_{ote}(\mathfrak{L}_a|\boldsymbol{d}, f) = \sum_f E_{ote}(\mathfrak{L}_b|\boldsymbol{d}, f) \tag{1}$$

The off-training-set error $E_{ote}(\mathfrak{L}_a|\boldsymbol{d}, f)$ is given in Eq.2, where $\mathbb{I}(\cdot)$ is 1 when $\cdot$ is ture, otherwise 0.

$$E_{ote}(\mathfrak{L}_a|\boldsymbol{d}, f) = \sum_h \sum_{d_i \in f, d_i \notin \boldsymbol{d}} P(d_i)\mathbb{I}(h(d_i) \neq f(d_i))P(h|\boldsymbol{d}, \mathfrak{L}_a) \tag{2}$$

## 3   THE TAFL THEOREM

We consider that a distribution can be parametrized as $f(: \theta)$. For a specific algorithm $\mathfrak{L}_a$, there is a set of distribution $\{f_a(: \theta_a)|\theta_a \in R\}$ which can be optimally solved by $\mathfrak{L}_a$. The TAFL Theorem is described as below.

> $\mathfrak{L}_a$ and $\mathfrak{L}_b$ are the optimal algorithms, in terms of off-training-set error, for distribution sets $\{f_a(: \theta_a)|\theta_a \in R\}$ and $\{f_b(: \theta_b)|\theta_b \in R\}$ respectively. $f_a(: \theta_a)$ has the form of $\boldsymbol{y} = g_a(\sum_i \boldsymbol{x}^i \theta_a^i + \theta_a^b)$. $g_a$ can be any continuous nonconstant function.
> Given a specific $f_b(: \theta_b^*)$ and its samples $\boldsymbol{d}_b^*$, we can find a serial of tasks, representing by a serial of sets of samples $\{\boldsymbol{d}_a^0, ..., \boldsymbol{d}_a^n\}$. $\mathfrak{L}_a$ is the optimal algorithm for each of $\boldsymbol{d}_a^i$ and if we solve each $\boldsymbol{d}_a^i$ in the provided order, then $E_{ote}(\mathfrak{L}_a|\boldsymbol{d}_a^n, f_b(: \theta_b^*)) = E_{ote}(\mathfrak{L}_b|\boldsymbol{d}_b^*, f_b(: \theta_b^*))$.

The proof is given as below.

> According to the universal approximation theorem Hornik et al. (1989), $f_b(: \theta_b^*)$ can be arbitrarily accurate approximated by a continuous nonconstant function. As $f_a(\boldsymbol{x} : \theta_a)$ has the form $\boldsymbol{y} = g_a(\sum_i \boldsymbol{x}^i \theta_a^i + \theta_a^b)$, the approximation has the form:
>
> $$\begin{aligned} \boldsymbol{y}_0 &= \boldsymbol{x} \\ \boldsymbol{y}_1 &= \{y_1^i|y_1^i = f_a(\boldsymbol{y}_0 : \theta_a^{1,i}), i = 0, ..., m\} \\ &\cdots \\ \boldsymbol{y}_n &= \{y_n^i|y_n^i = f_a(\boldsymbol{y}_{n-1} : \theta_a^{n,i}), i = 0, ..., k\} \end{aligned} \tag{3}$$
> $$f_b(\boldsymbol{x} : \theta_b^*) = \boldsymbol{y}_n$$
>
> Each mapping $f : \boldsymbol{y}_{m-1} \to \boldsymbol{y}_m$ has the form of $f_a(\boldsymbol{x} : \theta_a)$, therefore they can be optimally solved by $\mathfrak{L}_a$. But for each $f : \boldsymbol{f}_{m-\tau} \to \boldsymbol{f}_m, \tau \geq 2$, there is no guarantee that they have the form of $f_a(\boldsymbol{x} : \theta_a)$, therefore they may not be optimally solved by $\mathfrak{L}_a$.
> As each $f : \boldsymbol{y}_{m-1} \to \boldsymbol{y}_m$ has the form of $f_a(\boldsymbol{x} : \theta_a)$, they are also tasks generated by the distribution $f_a(: \theta_a)$.
> Hence, if a serial of tasks is provided in a order $\{f : \boldsymbol{y}_{m-1} \to \boldsymbol{y}_m|m = 1, ..., n\}$, $\mathfrak{L}_a$ can solve $f_b(: \theta_b^*)$ just as good as $\mathfrak{L}_b$.

The above proof does not conflict with the universal approximation theorem. The universal approximation theorem is not about the off-training-set error. Therefore, there is no guarantee that a feedforward network is optimal in terms of off-training-set error for any function.

One may misunderstand the serial of tasks is similar to hidden layers of a neural network. They are totally different concepts. Each task in the serial is sampled from the task distribution space. Hence, they can be considered as real applications. For example, one task is a OCR task, and another task is a translation task. But the hidden layers of a neural network are not related to real applications, only the outputs of the whole network are related to real applications.

## 4   ANALYSIS OF THE TASK ORDER

Next, we analyse the task order mentioned in the TAFL Theorem. We first describe our conclusion as below.

$\mathfrak{L}_a$ is the optimal algorithm in terms of off-training-set error for a set of distribution $\{f_a(: \theta_a)|\theta_a \in R\}$. $f_a(: \theta_a)$ has the form of $\boldsymbol{y} = g_a(\sum_i \boldsymbol{x}^i \theta_a^i + \theta_a^b)$.

Given a task of distribution $f_b(: \theta_b^*)$, $TO^* = \{\boldsymbol{d}_a^{0*}, ..., \boldsymbol{d}_a^{n*}\}$ is the correct task order for algorithm $\mathfrak{L}_a$ to optimally solve $f_b(: \theta_b^*)$. The task order $TO^*$ is a sample among all possible task order samples. For a new given task, we sample a set of task orders $TO^i$ randomly or following some distribution until the $TO^*$ is sampled. For each sampled $TO^i$, we call $\mathfrak{L}_a$ to solve each of $\boldsymbol{d}_a^i$ in it.

We declare that the number of calls of $\mathfrak{L}_a$ to solve each sampled $\boldsymbol{d}_a^i$ until $\boldsymbol{d}_a^{n*}$ is solved can decrease when the number of previously solved tasks increases.

Roughly speaking, we can learn how to combine models generated in previous tasks to solve new tasks to reduce the calls of $\mathfrak{L}$. The detailed reasons are listed below.

1. We can use a set $M$ to store models of all solved $\boldsymbol{d}^i$. Therefore, if a new task needs to solve a $\boldsymbol{d}^i$ which model is already in $M$, we can skip running $\mathfrak{L}_a$ on it.

2. We can construct a new kind of task based on $M$. This kind of task, denoted as abstract tasks, learns a mapping from $\boldsymbol{d}^i$ to a sub-set of models in $M$: $f_A : \boldsymbol{d}^i \rightarrow \boldsymbol{m}, \boldsymbol{m} \subset M$, where $\boldsymbol{m}$ should contain models of the correct task order for solving $\boldsymbol{d}^i$. The loss of this mapping is shown in Eq.4.

$$L(f_A|\mathfrak{L}_a, f_a, M) = \sum_{\boldsymbol{d}^i} \sum_{\boldsymbol{m}} E_{ote}(\mathfrak{L}_a|\boldsymbol{d}^i, f_a(\boldsymbol{m}(x) : \theta_a))P(\boldsymbol{m}|\boldsymbol{d}^i, f_A) \tag{4}$$

For a given $\boldsymbol{d}_b^*$, generated by $f_b(: \theta_b^*)$, if the model of a mapping $f_A$ can recognize it and give a set of models $\boldsymbol{m}^*$ which guarantees that $f_b(: \theta_b^*) = f_a(\boldsymbol{m}^*(x) : \theta_a)$, then $\mathfrak{L}_a$ can optimally solve this task directly. According to the TAFL Theorem, we know there must be an algorithm $\mathfrak{L}_A$ which is capable to optimally solve all possible $F_{abs} = \{f_A, f_B, ...\}$.

Each training sample of $f_A$ is a task in the distribution space $F$, and the model of $f_A$ is optimized to minimize off-training-set error. Therefore, the model of a single $f_A$ can be used to generate $\boldsymbol{m}$ for a set of tasks in $F$. It means after learning a single task $f_A$, we can decrease the number of running $\mathfrak{L}_a$ on a set of tasks in $F$.

Following this idea, we can also construct $f_{A^2} : (\boldsymbol{d}^i, \boldsymbol{m}) \rightarrow \boldsymbol{m}_{A^2}, \boldsymbol{m}_{A^2} \subset M_{A^2}$, where $M_{A^2}$ stores models generated in solving tasks in the distribution space $F_{abs}$. All possible $f_{A^2}$ form a new distribution space $F_{abs^2}$. The model of a single $f_{A^2}$ can be used to generate $\boldsymbol{m}_{A^2}$ for a set of tasks $f_A \in F_{abs}$ which decreases the number of running algorithm $\mathfrak{L}_A$ on those tasks.

Generally speaking, for the model of a task in distribution $F_{abs^i}$, it can be used to decrease the number of running algorithms $\mathfrak{L}_{A^{i-1}}$ on a set of tasks in $F_{abs^{i-1}}$. Please note, according to the TAFL Theorem, we can use the same algorithm for all $\mathfrak{L}_{A^i}$.

3. We consider situations that $f_{A^i} \in F_{abs^j}$ or $f_a \in F_{abs^j}$ for some $i$, $a$ and $j$. It means models of a distribution space can be used in another distribution space. We consider two situations.

In the first situation, $f_{A^i}$, from the distribution space $F_{abs^i}$, is a part of a task order $TO^{A^j}$. $TO^{A^j}$ is sampled for solving a task $f_{A^j}$ that comes from the distribution space $F_{abs^j}$. In this situation, the model of $f_{A^i}$ can be used directly to avoid running an algorithm for solving $f_{A^i}$ again.

In the second situation, $f_{A^i}$ is equal to $f_{A^j}$. Therefore, we can avoid learning $f_{A^j}$ and use the model of $f_{A^i}$ to solve tasks that $f_{A^j}$ are required to solve. In those two situations, the calls of learning algorithms are decreased.

The elements $\boldsymbol{m}$ of inputs and outputs of $f_{A^i}$ and $f_{A^j}$ are from different model sets. One may argue that how $f_{A^i} \in F_{abs^j}$ is possible. Here we suggest some simple ways. For example, we can put all models in the same set, or we can use an encoding function to map any $\boldsymbol{m}$ to a real-valued embedding $E : \boldsymbol{m} \rightarrow \boldsymbol{r}$. An example is given in the section.5 which shows a way to allow $f_{A^i} \in F_{abs^j}$.

## 5 EXAMPLE OF THE TAFL THEOREM

We present a simple example to help readers to understand the TAFL Theorem. Please note, we are not going to prove this example is better than any existing algorithm.

We generate a serial of 2-dimensional binary classification tasks using several task distribution listed in Eq.5. The $f_s$ of those distribution is a 2-dimensional affine transformation followed by a sigmoid function, as shown in Eq.6. The distribution in $\{f_i|i = 0, ..., n\}$ can be arbitrarily accurate approximated by $f_s$ directly, but $\{f_i|i = n + 1, ..., m\}$ can not. With strategies mentioned in the section.4, we are going to use $f_s$ to approximate all $\{f_i|i = 0, ..., m\}$.

$$
\begin{aligned}
&f_0 : f_s(\boldsymbol{x} : \theta_0 + \tau) \\
&... \\
&f_n : f_s(\boldsymbol{x} : \theta_n + \tau) \\
&f_{n+1} : f_s((f_i, f_j) : \theta_{n+1} + \tau), i, j \in 0, ..., n \\
&... \\
&f_m : f_s((f_k, f_q) : \theta_m + \tau), k, q \in 0, ..., n
\end{aligned}
\tag{5}
$$

$$
f_s(\boldsymbol{x}) = \frac{1}{1 + e^{-\sum_i x_i \theta_i + \theta_b}}
\tag{6}
$$

Each task randomly chooses a task distribution $f_i$ and randomly samples $\tau$. The $\{\theta_i|i = 0, ..., m\}$ are fixed across all tasks. The inputs of a task $i$ are uniformly sampled from $[0, 1]$, denoted by $X_i = \{(x_i^0(k), x_i^1(k))|k = 0, ..., m\}$. The targets are given by $Y_i = \{\mathbb{I}(f_i(\boldsymbol{x}_i(k)) > 0.5)|k = 0, ..., m\}$. We show some tasks in Fig.1, where different colors represent different classes. The subfigures $a$ to $e$ are from distribution $\{f_i|i = 0, ..., n\}$, and $f$ to $h$ are from distribution $\{f_i|i = n + 1, ..., m\}$.

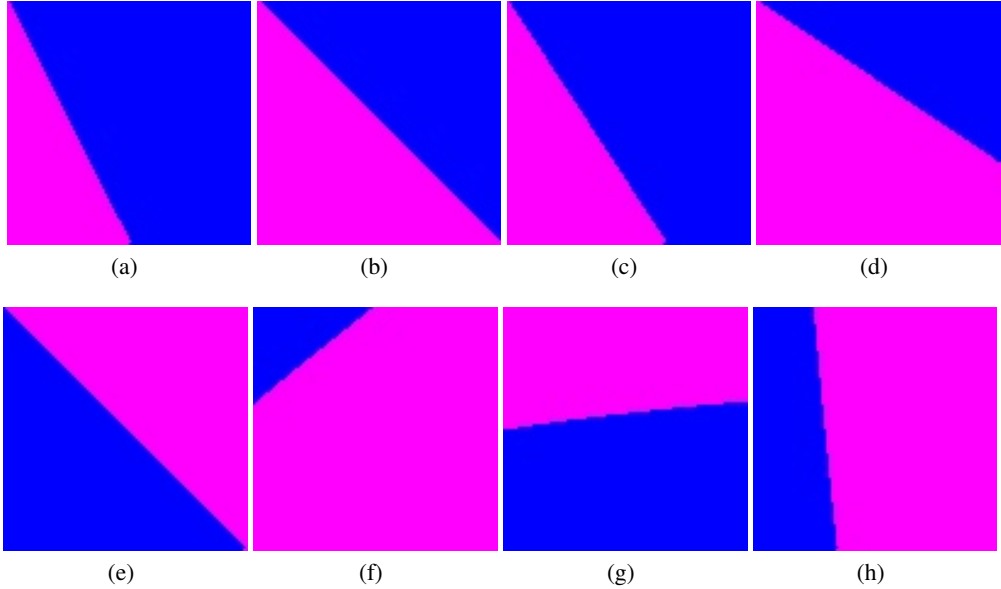

Figure 1: 2-dimensional binary classification tasks

Given a task $T_i$, we first try to use $f_s$ as the model to solve this task. If the resulting accuracy is above a threshold $\theta_{acc}$, we store the model of this task, denoted as $m_i$, in a set $M$. If the accuracy is below the threshold $\theta_{acc}$, we use random search to choose two models $m_k, m_q$ in $M$ to form a new model $f_s((m_k(x), m_q(x)) : \theta)$ to replace $f_s$ and run from the first step again. We terminate this task when we get a qualified model or the number of runs exceeds a threshold $\delta$. This process is marked as $\mathfrak{L}$.

Besides tasks generated from $f_i$, we also generate abstract tasks to learn how to choose models for new tasks. For two random given $m_k, m_q$ in $M$, we randomly generate two new models $m_0 = f_s((m_k, m_q) : \theta_0)$ and $m_1 = f_s((m_k, m_q) : \theta_1)$. Next, we generate some classification tasks using $f_s((m_k, m_q) : \theta_0 + \Delta\tau)$ and $f_s((m_k, m_q) : \theta_1 + \Delta\tau)$ as task distribution. We also generate some classification tasks using arbitrary $f_i$. We calculate the test accuracy of generated tasks on both $m_0$ and $m_1$. The test accuracy of those tasks forms a set of training inputs $X = \{(e^0(z), e^1(z)) | z = 0, ..., m\}$. We use whether a task is generated by $f_s((m_k, m_q) : \theta_0 + \Delta\tau)$ or $f_s((m_k, m_q) : \theta_1 + \Delta\tau)$ as training targets. Those samples form a new abstract task $T_j$. We use the above process $\mathfrak{L}$ to solve this task.

After solving some abstract tasks, the step of selecting $m_k, m_q$ in $\mathfrak{L}$ can be improved. We first check whether the given task can be recognized by models of abstract tasks in $M$. If so, we use the corresponding $m_k, m_q$ directly. In addition, if a model in $M$ can achieve a relatively higher accuracy on $T_i$, we use this model to initialize $\theta$ of $f_s(: \theta)$.

The whole process in shown in Algorithm.1 and Algorithm.2.

---

**Algorithm 1** Solving a task

---

**Require:** task distribution space $F$ and model $f_s(x : \theta)$
  Set $M = \{\}$
  **while** not end **do**
    generate a task $T_i$
    **if** $f_s$ can solve $T_i$ **then**
      add new model $m_i$ into $M$.
    **else if** $M$ can recognize $T_i \rightarrow m_k, m_q$ and $f_s((m_k, m_q) : \theta)$ can solve $T_i$ **then**
      add new model $m_i$ into $M$.
    **else if** here is a $m_k \in M$ and $f_s(: \theta_k + \Delta)$ can solve $T_i$ **then**
      add new model $m_i$ into $M$.
    **else**
      **while** $runs < \delta$ **do**
        **if** randomly choose $m_k, m_q$ and $f_s((m_k, m_q) : \theta)$ can solve $T_i$ **then**
          add new model $m_i$ into $M$.
        **end if**
      **end while**
    **end if**
  **end while**

---

**Algorithm 2** Generating a task

---

**Require:** $M$ and task distribution space $F$.
  **if** sample a task from $F$ **then**
    **return** a task randomly generated from $F$.
  **else**
    randomly choose $m_k, m_q$ form $M$.
    randomly generate two models: $m_0 = f_s((m_k, m_q) : \theta_0)$ and $m_1 = f_s((m_k, m_q) : \theta_1)$
    generate $N$ classification tasks using $f_s((m_k, m_q) : \theta_0 + \Delta\tau)$ and $f_s((m_k, m_q) : \theta_1 + \Delta\tau)$
    generate $M$ classification tasks from $F$
    calculate test accuracy of those tasks on $m_0$ and $m_1$.
    **return** $X = \{(e^0(i), e^0(i)) | i = 0, ..., N + M\}, Y = \{\mathbb{I}(i < N) | i = 0, ..., N + M\}$
  **end if**

---

We use SGD as a learning algorithm to solve each task. We measure task failure rate and the runs of SGD of a single task in a sliding window manner and show them with the number of tasks in Fig.2. We also measure how a task is solved and show it in Fig.3. The "fail" and "fs" of the Fig.3 are the percentage of failed, directly solved tasks respectively. The "random" and "recognize" mean a task solved by randomly selecting $m_k, m_q$, and recognizing $m_k, m_q$ respectively. The two figures show that the failure rate and the number of runs both decrease with the number of solved tasks increasing. The reason of this phenomenon is more and more tasks are solved in the recognizing way as shown in Fig.3.

In conclusion, this example shows a possible implementation of the TAFL Theorem, and demonstrates our main ideas: it is possible to use an algorithm to learn that it can not directly learn. The more we learn, the faster we solve new tasks.

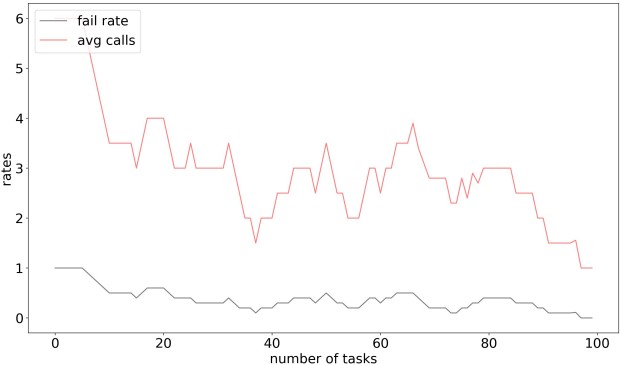

Figure 2: Failure rate and runs of SGD in a single task.

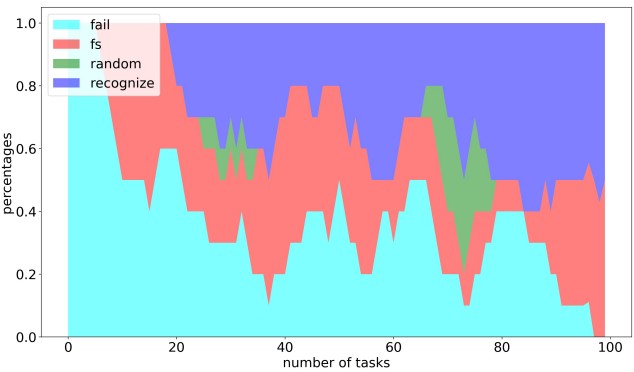

Figure 3: Percentages of how a task is solved.

## 6 IMPLICATIONS OF THE TAFL THEOREM

The idea behind the TAFL Theorem is very simple. One only has limited ability to directly solve unknown tasks. For a task one cannot directly solve, the prior skills of this task may be directly learnable. One should learn those prior skills first and then learn how to use those skills. After that, the unsolvable task becomes directly solvable. Maybe even human intelligence is not powerful enough to solve complex tasks directly. On the contrary, maybe, human intelligence can only solve relatively simple tasks directly. But with some simple strategies that ensure sharing knowledge across tasks, all complex tasks can be converted into simple tasks that human intelligence can solve directly.

The main methodology of the current AI community is to design different algorithms for different AI problems to ensure the problems can be solved directly. This methodology can converge on all possible problems only if all possible problems can be solved by finite algorithms. We are very doubtful about this assumption. The alternative way is to keep algorithms fixed, but to use solved tasks to improve the ability of solving new tasks. The disadvantage is some tasks can not be solved at the first time. We need to wait until the whole system has the ability to solve this task. We believe this methodology is more likely to converge on all possible problems.

The TAFL Theorem implies two kinds of applications. The first kind of application uses models generated earlier to improve performance of current tasks. We have already seen many methods that utilized this idea. Some methods directly use outputs of another model to improve the performance of current tasks. In 3D detection, FusionPainting Xu et al. (2021) feeds outputs of a fixed 2D segmentation model to a 3D detection network to largely improve the accuracy of 3D detection. In image captioning Wang et al. (2020), it is very common to use a fixed classification or detection model to provide information of the image. The information is fed into another network to generate image captions.

Transfer learning can be considered as a variant of this idea. For example, it is very popular to use pre-trained models in downstream tasks, e.g. ImageNet pre-trained models Neyshabur et al. (2020) in computer vision and BERT Devlin et al. (2018) in natural language processing (NLP).

Task adaptation is also related to this idea. MAML Finn et al. (2017) proposes a framework to train a meta-model on source tasks and fine-tuning it on target tasks. The meta-model should be capable of adaptation easily to target tasks. It is done by simulating the adaptation process many times, and optimize the meta-model to maximize the adaptation ability across those simulations.

The second kind of application learns a model to generate a combination of old models for a given task. The combination is used in the same way as the first kind of application to improve performance of the given task. To our knowledge, there are few works about this idea. Maybe, this is because solved tasks at our hands are not enough or properly organized.

Some ensemble methods can be considered related to this idea. For example, CollaborationofExperts Zhang et al. (2021) archives 80% Top-1 Accuracy on ImageNet with 100M FLOPs by generating multiple models in training and selecting the most appropriate one for predicting.

## 7    CONCLUSION

This paper proposes the TAFL Theorem which indicates that with some simple strategies, some algorithms are capable to optimally solve all tasks. Base on the TAFL Theorem, we also conclude that with the number of solved tasks increasing, the algorithm can solve new tasks faster.

The TAFL Theorem also implies that maybe it is not a matter that algorithms at our hands are not powerful enough. An alternative way is to accumulate models of solved tasks and learn how to use those models to solve new tasks.

This paper presents an example to demonstrate the TAFL Theorem. The initial inductive bias is defined as a 2-dimensional affine transformation followed by a sigmoid function. The tasks to solve are a serial of 2-dimensional binary classification tasks. Some of them can be directly solved by the initial inductive bias, but some can not. With a learning strategy implied by the TAFL Theorem, the example shows both the failure rate and the runs of a learning algorithm decrease with the task number increasing.

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
