# OpenReview forum: "There are free lunches"
_ICLR.cc/2022/Conference — ICLR 2022 Submitted_

### Official Review · Reviewer_BuWJ · 2021-10-20

**Correctness:** 1
**Technical Novelty And Significance:** 4
**Empirical Novelty And Significance:** 1
**Recommendation:** 1
**Confidence:** 3

**Main Review:**

Note:Since i am unfamiliar with the notation and some of the terms used, i see a possibility for error in my review and lower my confidence as a result of this.

I believe the statement is either wrong or trivial.

From a philosophical standpoint, there is a very intuitive reason for this:
Lets assume a meta-algorithm L_m that given D for task T

a) finds a task split

b) for each intermediate task selects a distribution and sampling strategy from which to draw the new labels

c) finds the optimal model to solve the task using L_a.

Then, the NFL theorem applies to L_m and there must exist a L_b with problem class C for which  L_m is dominated by L_b for all T in C, i.e. not optimal in the sense of out-of-sample-error. Even if the proposed theorem is correct, L_m is only expanding the set of tasks L_a can solve optimally, but the NFL then implies that there must be a set of tasks it can't solve (and even worse than L_a due to the equal averages of NFL)

Moreover, I think that the theorem itself is wrong. Unfortunately the analysis of the paper does not take into account the accumulated composition errors of each task T_i. Even if L_a can solve each T_i optimally (in the sense of minimizing the out of sample error for this task), there is no guarantee that the final composition of the models returned by the meta strategy L_m has out-of-sample error on the same order as L_b would produce. The proof does not cover this aspect and assumes instead optimality of the composition.


Smaller things:
- The notation is poorly explained, e.g. the second paragraph of section 2 is quite unclear (D not defined, x/y and their relation to D only implicitly defined, and S is very poorly define given that there are quite a few choices for distribution spaces with different properties)

- In the TAFL theorem, f_a should probably be defined more precisely. E.g., the sum should have an upper bound or requirements on the bounds should be given (e.g., i assume it should be finite).

- It is unclear to me as i am no expert on generalized functions, but is it allowed to pick g_a=Id and the upper bound of the sum as 1? Would the theorem still hold? Or do you need some approximation properties between S and the model class f_a(:theta)?

**Summary Of The Paper:**

The paper introduces the TAFL theorem "There are free lunches" proposing a way around the No-Free-Luch theorem. The idea is to split a learning task T that can be solved optimally by a learning algorithm L_b into a set of N learning tasks T_i. The tasks T_i are chosen in such a way that a  learning algorithm L_a (which can not solve T optimally) can solve each T_i optimally in the sense that the predicted model h_i minimize the out-of-sample-error on T_i given dataset D_i. The proposition is then that the composition of the learned models h_i(h_2...h_N(x))..) is equivalent to a model h^* found by L_b for T in the sense that it minimizes the out-of-sample-error.

The optimal task decomposition (or task order) is analyzed as well.

**Summary Of The Review:**

The theorem is either trivial (it is impossible to find an optimal task split) or wrong (solving the task in the task-split optimally does not mean solving the original task optimally).

---

> ### Author Response · Authors · 2021-11-11
> **Responses to Reviewer BuWJ**
>
> We are sincerely grateful for your time and efforts in the review process.
>
> 1. For your questions:
>
> """
>
> Lets assume a meta-algorithm L_m that given D for task T
> a) finds a task split
> b) for each intermediate task selects a distribution and sampling strategy from which to draw the new labels
> c) finds the optimal model to solve the task using L_a.
> Then, the NFL theorem applies to L_m and there must exist a L_b with problem class C for which L_m is dominated by L_b for all T in C, i.e. not optimal in the sense of out-of-sample-error.
>
> Even if the proposed theorem is correct, L_m is only expanding the set of tasks L_a can solve optimally, but the NFL then implies that there must be a set of tasks it can't solve (and even worse than L_a due to the equal averages of NFL)
>
> """
>
> Our explanations:
>
> We think you mean 'L_b' is another meta-algorithm and 'L_b' is better than 'L_m' on finding task splits on some tasks.
> Please consider how do we get 'L_b' or 'L_m'. There must be a task 'T_b'. We solve this task and get 'L_b'. In the same way, we solve another task 'T_m' and get 'L_m'.
> We know both 'T_b' and 'T_m' can be split into sub-tasks which can be optimally solved by 'L_a'. In this way, there are specific task splits to let 'L_a' optimally solve 'T', 'T_B' and 'T_m'.
> In other words, it is possible to use one algorithm to generate all other algorithms. This is why TAFL is not limited by NFL.
>
> Next, we do not know the optimal task splits for tasks 'T_b' and 'T_m'. In the worst case, we randomly generate task splits with the hope that the optimal task splits are finally sampled. In the best case, we already have another algorithm 'L_c' which can generate the optimal task splits for 'T_b' and 'T_m'.
> One may think that to get 'L_c', there must be an algorithm 'L_d'. To get 'L_d', there must be an algorithm 'L_e'. This leads to an infinite chain of algorithms.
> Fortunately, as we analysed in Section.4, following some strategies and assumptions, we can both avoid randomly sampling task splits and break this infinite chain.
>
>
> 2. For your questions:
>
> """
>
> Moreover, I think that the theorem itself is wrong. Unfortunately the analysis of the paper does not take into account the accumulated composition errors of each task T_i. Even if L_a can solve each T_i optimally (in the sense of minimizing the out of sample error for this task), there is no guarantee that the final composition of the models returned by the meta strategy L_m has out-of-sample error on the same order as L_b would produce. The proof does not cover this aspect and assumes instead optimality of the composition.
>
> """
>
> Our explanations:
>
> The accumulated composition error is indeed a problem in practice. But in theory, if L_m is optimal, there is no accumulated composition error. This is guaranteed by universal approximation theory.

---

> > ### Comment · Reviewer_BuWJ · 2021-11-11
> > **The second answer misses the point**
> >
> > Hi,
> >
> > regarding point 2:
> > The universal approximation theory only states: there exists a model h^* in the set of models that approximates the target function arbitrarily well. It is not considering data and is therefore only relevant to answer the question: can the model class solve the problem at all?
> >
> > But due to the fact that the NFL does consider the finite data case, the dataset might not contain enough information for L_m to identify h^*. Indeed, Frequentist theory states bounds on the precision of finding said model given a limited amount of data, e.g., the Cramér–Rao bound.
> >
> > Thus, accumulated composition error is not only a problem in practice, but also in theory.

---

> > > ### Author Response · Authors · 2021-11-11
> > > **Responses to Reviewer BuWJ**
> > >
> > > Thanks for the helpful comments. We analyze this problem as below.
> > >
> > > Lets assume the error of algorithm 'L_a' is 'e_a' on distribution set 'D_a'.
> > >
> > > 'd_1'  is a task form 'D_a'. We solve it using 'L_a' and get a model  'f_1' with error 'e_a'.
> > >
> > > 'd_2' is a task from another distribution set 'D_b'. The inputs and targets of 'd_2' are denoted as 'x' and 'y'.
> > >
> > > Let 'o=f_1(x)', there are two situations:
> > > 1. 'o->y' belong to 'D_a'. We can use 'L_a' to solve it with error 'e_a'. This is no accumulated composition error.
> > > 2. 'o->y' do not belong to 'D_a'. We can not use 'L_a' to solve it or with accumulated composition error.

---

> > > > ### Comment · Reviewer_BuWJ · 2021-11-11
> > > > **Third case is missing**
> > > >
> > > > Hi,
> > > >
> > > > as far as i understood your paper, your proposal would not consider the case 2, but instead a case 3 where you would then proceed to generate a new task sequence d_21,d_22...d_2n, each solvable by L_a with error e_21,e_22 ... e2n. And I say: there is no guarantee that the final composition of models has smaller error than solving the task directly with L_b.
> > > >
> > > > If yes, please provide the part in your manuscript, where you show that.

---

### Official Review · Reviewer_SgLo · 2021-11-02

**Correctness:** 2
**Technical Novelty And Significance:** 2
**Empirical Novelty And Significance:** Not applicable
**Recommendation:** 3
**Confidence:** 3

**Main Review:**

+   Overall,   the   paper   is   well   written   in   terms   of   structure.   Also,   the   problem
statement is interesting and there can be many powerful works in this area.
+ No grammatical errors or typos found in the paper.

Concerns:
- The key concern about the paper is that the assumptions made do seem strong, and
this is because the ordering of tasks and solving instances in a particular condition
is only applicable in few contexts..
- There are doubts in accepting the formulation because the proof is not clear, and
the example stated in the paper is trivial; no experiments are done to
prove the formulation. Also, the conclusions made about the ordering of task seems
still unclear.
- Another concern with the paper is the lack of a background section discussing
different things stated in the paper. There is not much background discussion about
the NFL theorem except the main paper.
- Also, it would be nice if there was a piece of information about mathematical
notations like the off-training-set error stated in the paper, and also the universal
approximation   theorem   which   the   proof   of   the   formulation   is   inspired   by   this
theorem but there is no discussion about that.
- This lack of background section and some confusing denotations in the equations
with no explanation for them makes the paper hard to follow. I am not sure if I have
missed points while understanding the equations since some notations were unclear
and confusing.
- In equation 3, it is not clear how we can reach the point that if tasks are in specific
order algorithm a can solve the b's distribution as optimal as algorithm b. I would
explain it more to make it more convincing.
- It is not clear enough for me to accept the statement that the number of calls of
the   algorithm   would   decrease   when   the   number   of   previously   solved   tasks
increases.
Minor Comments:
- Figure 3's y-axis is mentioned to be in percentage but it's in the [0,1] range.

**Summary Of The Paper:**

The paper provides a new optimization theorem called "There Are Free Lunches"
which in particular argues the NFL theorem. Based on the assertion, although the
NFL states that averaged overall data distribution all algorithms have the same
error rate, the TAFL theorem proposes that some algorithms can achieve the best
performance if tasks are given in special order. Also, it is declared that if more
problems are solved the difficulty of the new task will be decreased. They also
mentioned an example to use the theorem. The proof of the TAFL theorem is
inspired by the universal approximation theorem. The main contribution of the
paper is to formulate the theorem and bring one example for the theorem.

**Summary Of The Review:**

Unclear and unconvincing.

---

### Official Review · Reviewer_WGUc · 2021-11-02

**Correctness:** 3
**Technical Novelty And Significance:** 3
**Empirical Novelty And Significance:** 2
**Recommendation:** 5
**Confidence:** 2

**Main Review:**

After reading the manuscript I was left with lots of questions that undermined my capacity to evaluate its correctness and impact. I list the main questions below in the hope that each question provides an opportunity for the authors to improve the readability and notation of the paper:

- Is the theorem concerned only with learning algorithms or algorithms with memory or is it also applicable to general optimization methods? Please state more clearly the differences in scope between TAFL and NFL.
- In ${f_a (: \theta_a )|\theta_a ∈ R}$ is $\theta_a$ a real number, a vector? If so, use the adequate symbol for the real set.
- In $y = g_a (\sum_i x^i \theta_a^i + \theta_a^b )$, why the superscript $b$ in $\theta^b_a$? Is it connected with $f_b$? If so, please, explain how and why this is necessary.
- In Eq. 2, $E_{ote}(L|d, f)$ is defined with $\bf d$ sampled from $f$, since $d_i$ ∈ $f$ but in TAFL Theorem we have  $E_{ote}(L|d_a, f_b)$. How can $d_a$ be sampled from $f_b$? And how evaluating the performance of $L_a$ in $d_a^n$ can say anything about the performance of $L_b$ in $d_b^*$? This connection is not clear to me even after reading the proof provided.
- In Eq 3, how $y_0$ can be equal to $x$? Does that imply certain assumptions about the mapping $f:x->y$? If so, what assumption? Similarly, why $f_b (x : \theta_b^∗ ) = y_n$? What are the assumptions about $d$? Do $x$ and $y$ come from the same domain? For instance, in a binary classification, how can $x$ and $y$ domains match? What about $d_a$ and $d_b$?
- What does $n$, $m$ and $k$ mean in Eq. 3?
- In Eq. 5, what assumptions allow the composition of $f_s$ with $f_i$ and $f_j$? In other words, why can $(f_i, f_j)$ replace $x$?
- In Section 2 $h$ is used to represent the learned model. In Section 4 $m$ is used. Is there a reason for the change? What is the difference between bold *$m$* and $m$?

Improve presentation:
- Please, review the sentence: “we can find a **serial** of tasks, represen**ting** by a **serial** of sets of samples”.  "series of tasks" or "sequence of tasks" would fit better in my opinion. Overall, the paper could benefit from an English revision.
- Citations in the text should be fixed to use outer parentheses when the author’s names do not belong to the sentence.
- Fix and standardize reference to textual elements: section.5 -> Section 5.


**Summary Of The Paper:**

The paper addresses an important topic in the theory of learning algorithms which regards the limits of learning algorithms. If correct, the theorem provided in the paper could have a strong theoretical impact on the field of machine learning and maybe also practical applications.

**Summary Of The Review:**

Considering the difficulty to understand the proof presented in the article, I do not think the paper is ready to be accepted in its present form. I can reconsider this position if the needed clarifications and improvements are provided.

---

### Official Review · Reviewer_tzHo · 2021-11-02

**Correctness:** 2
**Technical Novelty And Significance:** 2
**Empirical Novelty And Significance:** Not applicable
**Recommendation:** 3
**Confidence:** 2

**Main Review:**

The paper is not well written and the mathematical notation in the expression of the theorem Is very poor and there are definitions that are missing. For example, the \theta sometimes appears only with a subscript index and sometimes with a subscript and superscript. Also the "f(:\theta)" is also confusing. What is the purpose of the colon here? It is also unclear whether \theta is a vector or a scalar. Also in equation 3  n,m,k are undefined. At last, the term serial tasks is probably not correct. task sequence would have been more appropriate.
Given the above problems in the notation, I wasn't able to follow the proof. I also wanted to criticize the approach to the problem which seems very abstract, very theoretical, and most likely does not seem to be useful. There are other approaches that although answer the same problem in a more practical manner. The recent paper "Efficiently Identifying Task Groupings for Multi-Task Learning" https://arxiv.org/pdf/2109.04617.pdf is very closely related to the problem posed here and provides a practical algorithm for solving the problem. There is a difference between the two problems. The paper under review claims that there is a sequence while the other paper trains them together.

At last, one of the things that the authors should have used in their paper is a connection to catastrophic forgetting. Catastrophic forgetting is a phenomenon that is caused by the sequential learning of tasks. It would have been nice to show how this is addressed by TAFL.


**Summary Of The Paper:**

The paper is going beyond the No free lunch theorem and tries to answer the question: "Is it possible to reorder tasks so that I can learn them better?" The paper claims that there is an optimal sequence of task training, but it doesn't provide an algorithm for finding it. There are no real-world or nontrivial synthetic data experiments. One way of understanding the question of the paper is the following example:
Consider a transformer network. We know empirically that training a task that predicts the next word and then a machine translation gives good results. Is that the optimal order of training these tasks?

**Summary Of The Review:**

I don't claim to be an expert in ML theory, but given the simplicity of the concepts used in the proof, it should have been easy to follow. This is not the case. An opinion from a theory expert should be more appropriate. It is my impression that the reviewers that would be capable of giving a clear opinion are probably not at ICLR. This paper should be submitted to venues like COLT or SODA. The paper needs more work and more clarity in the notation. My recommendation is to reject the paper.

---

### Decision · Program_Chairs · 2022-01-20

**Decision:**

Reject

**Comment:**

As the reviewers say, the subject matter of this paper is important, and of interest to the ICLR audience (I discount tzHo's suggestion that the paper is more suited to other venues).

However, there are three primary reasons this paper should not be published as is:
1. A theoretical paper *must* be precise, accurate, and clear.
  The reviewers universally consider the notation ambiguous, and the theorem unproved because of this ambiguity.
 2. The leap from solving a series of tasks optimally to having solved the composition optimally is indeed poorly argued in the paper, and is not resolved by the discussion.
 3. I would also strongly recommend showing a less trivial example.  It does not need to be "real world", but it should address numerically the specific doubt of BuW: the relationship of OTE of the composed model to OTE of the subtask models.

In summary: TaFL may be true, but this paper does not show it to be true; or conversely, TaFL may be false, in which case publishing this paper would be a grave error.  The authors should use the reviewers reports to clarify and strengthen the argument.  This does include showing numerical results, because inspection of the code generating such results can often aid reviewers and readers in judging the truth of the theoretical claims, and in finding subtle missteps in the derivations.